# Validation of a LLME/GC-MS Methodology for Quantification of Volatile Compounds in Fermented Beverages

**DOI:** 10.3390/molecules25030621

**Published:** 2020-01-31

**Authors:** Eduardo Coelho, Margarida Lemos, Zlatina Genisheva, Lucília Domingues, Mar Vilanova, José M. Oliveira

**Affiliations:** 1CEB—Centre of Biological Engineering, University of Minho, 4710-057 Braga, Portugal; e.coelho@ceb.uminho.pt (E.C.); margarida.lemos.20@gmail.com (M.L.); zlatina@deb.uminho.pt (Z.G.); luciliad@deb.uminho.pt (L.D.); 2Spanish National Research Council (MBG-CSIC), Salcedo, 36143 Pontevedra, Spain

**Keywords:** fermented beverages, volatile compounds, analytical method, liquid-liquid microextraction, GC-MS

## Abstract

Knowledge of composition of beverages volatile fraction is essential for understanding their sensory attributes. Analysis of volatile compounds predominantly resorts to gas chromatography coupled with mass spectrometry (GC-MS). Often a previous concentration step is required to quantify compounds found at low concentrations. This work presents a liquid-liquid microextraction method combined with GC-MS (LLME/GC-MS) for the analysis of compounds in fermented beverages and spirits. The method was validated for a set of compounds typically found in fermented beverages comprising alcohols, esters, volatile phenols, and monoterpenic alcohols. The key requirements for validity were observed, namely linearity, sensitivity in the studied range, accuracy, and precision within the required parameters. Robustness of the method was also evaluated with satisfactory results. Thus, the proposed LLME/GC-MS method may be a useful tool for the analysis of several fermented beverages, which is easily implementable in a laboratory equipped with a GC-MS.

## 1. Introduction

The flavor, which is one of the most important sensory attributes of fermented, alcoholic, and distilled beverages (cider, wine, beer, vinegar, spirits, vodka, whiskey, among others), is determined by a vast and diverse number of volatile compounds, arising either from raw material (e.g., grapes, barley, hops), yeast/bacteria fermentations, which are secondary metabolites [1,2,3], or from ageing when applied (e.g., in oak wood) [4]. These volatile compounds belong to diverse chemical families like alcohols, esters, aldehydes and carbonyls, volatile fatty acids, volatile phenols, sulphur compounds, terpenes, norisoprenoids, lactones, furans, and more [3,5,6,7], which are often found in very low concentrations.

Since volatile compounds of fermented/alcoholic beverages are highly correlated with the sensory characteristics of the products, its identification and quantification acquires crucial significance for understanding beverages organoleptic properties and further develop product quality. In addition, the presence/absence or the amount of each individual component may be a marker of the used technology or the indication of a product defect. The analysis of individual volatile compounds must comprise a chromatographic separation, which is followed by a generic or a selective identification (e.g., flame ionization detector, electron capture detector, flame photometric detector, mass spectrometric detector) [8]. Recently, some authors have correlated FTIR spectra with some specific compounds or groups of compounds [9,10].

Apart from the major volatile compounds present in amounts of mg/L (e.g., 3-methyl-1-butanol), which may be analyzed by direct injection, those presented in lower amounts ranging from a few µg/L (e.g.*,* linalool) or even scarce ng/L (e.g.*,* 4-methyl-4-mercapto-2-pentanone) must be concentrated before the chromatographic separation. This step could be achieved by mixing a solvent with the sample, as in liquid-liquid extraction (LLE) [11,12] and liquid-liquid microextraction (LLME) [13,14]. Several adaptations/modifications of LLE/LLME methods can be envisioned, e.g., the evaporation of solvent for increasing concentration, adsorption of volatiles in a solvent drop (single-drop microextraction – SDME) [15], or even adsorption/desorption of the compounds using a polymeric phase (sorbent-phase extraction – SPE) [16]. Solvent-free techniques include solid-phase micro-extraction (SPME) [17,18], usually in the headspace of the sample (HS-SPME) [19,20], and stir-bar sorptive extraction (SBSE) [21,22]. Some of these methods, developed to analyze volatiles in alcoholic/fermented beverages, are generic considering that they allow the identification and quantification of the majority of compounds, where their range of application depends on the solvents and/or the sorbents’ polarity. Specific methodologies based on polymeric materials, sometimes applying derivatization procedures, were developed to quantify specific compounds or classes of compounds [23].

For a method to be applied in the laboratory, it must be validated to ensure its reliability and the quality of the obtained results. Several points must be addressed for a method to be valid, namely its linearity, specificity, quantification range, limits of detection and quantification, sensitivity, precision, and accuracy. Optionally, robustness and reproducibility studies can be performed to reinforce the methods applicability and efficiency [24,25,26].

This work aims to validate a liquid-liquid microextraction method (LLME) first published by Oliveira and collaborators [27], which only reported its use for the analysis of three C6-alcohols (1-hexanol, *E*-3-hexenol and *Z*-3-hexenol), exclusively in wine. As the method provided satisfactory performance and results, its feasibility for the analysis of a broader range of compounds and a wider variety of matrices remained to be validated. The presented LLME method combined with GC-MS poses as an additional alternative to analyze volatile compounds in alcoholic/fermented beverages. This procedure can be applied in any laboratory equipped with a GC-MS by any technician, using only ordinary glassware and low amounts of sample and solvents. High throughput applications are envisioned as the procedure enables handling a substantial number of samples and screening a large number of volatiles in a short period of time.

## 2. Results and Discussion

### 2.1. Linearity and Sensitivity

Linearity and sensitivity of the proposed LLME method were evaluated by outlining calibration curves for each analyte, using a solution of pure standards. Compounds selected for calibration of the method were chosen on the basis of their contribution for the volatile and aromatic fraction of fermented products, which are considered to be representative of the analytes generally found in beer, wine, spirits, and vinegar. Acids were left out of the validation study by considering the difficulties of maintaining them in a standard solution due to their reaction with some other components in the mixture.

Regressions were performed from the obtained data with the corresponding coefficients presented in Table 1. Good linear regressions were obtained for extraction and quantification using the LLME method, with values of *R*^2^ > 0.995 for all of the studied analytes. The *R*^2^ value is a useful indicator of the regression quality. However, according to Kruve and collaborators [26] and Araújo [28], it cannot be considered as a standalone measure to validate a method linearity, which must be further validated by a statistical lack-of-fit *F*-test. Lack–of-fit tests were performed for the regression curves obtained for each analyte, according to the recommendations of Araujo [28], since all regressions were demonstrated to be linear, with the *F* obtained being lower than the tabulated one for the corresponding degrees of freedom. This linearity reflects not only the directly proportional response of the MS detector, but also the direct proportionality in the extraction of analytes by LLME.

Extraction selectivity was maintained throughout the tested concentrations, which enabled proper quantification of the analytes in the mixture. All regressions presented intercept values not significantly different from zero (*p* > 0.05) and, therefore, equations are only based on the slope, similarly to the previously reported works for other LLE methods [12]. Moreover, the baseline value is subtracted for the integration of peaks in the chromatogram using background correction in the software, which also justifies the absence of the intercept value.

Sensitivity is defined as the change in the method response, which corresponds to a change in the measured quantity and is intrinsically related to the slope of the calibration curve [29]. In this case, $R_{f}$ is the inverse of the slope. This factor is, therefore, a measure of the method’s sensitivity in terms of the relative response of each compound in relation to the response of the internal standard. A higher response factor means a higher variation of the compound’s concentration for a given variation of the signal, which, therefore, accounts for a lower sensitivity. Overall, response factors obtained for esters ranged between 1 and 2, with the exception of ethyl lactate for which the response factor was highly superior while attaining the value of 44.9 and accounting for a lower sensitivity of the method toward this compound. Similar to esters, monoterpenic alcohols as well as 4-methyl-2-pentanone presented response factors between 1.2 and 1.8, which was followed by volatile phenols that presented slightly higher $R_{f}$ values of about 2. With higher response factors, and, therefore, lower sensitivity alcohols, presented $R_{f}$ values between 3.6 and 5.1. This variation in the response factor is a combination of different extraction selectivity by the LLME method and differences in ionization and detection in the MS. Response factors seem to be similar within groups of compounds, which, despite not excluding the need for determining a specific analyte response for a proper quantification, can aid in the prediction of the response for compounds within the same group. With response factors between 1 and 5 for the majority of compounds, it is believed that the method has good sensitivity. Therefore, the method complies with the first base requirements for validation, being that the LLME method in the study presents proper sensitivity and linearity.

### 2.2. Limits of Detection and Quantification

Limits of detection (*LOD*) and quantification (*LOQ*) deal with the minimum amount of compound possible to detect and quantify, respectively. As stated by Brettell and Lester [30], two strategies can be used for determining *LOD*: a statistical approach, which is more likely to generate artificially low *LOD* values, and an experimental approach, which is attained by decreasing the analyte concentration until the identification criterion is no longer met. This generates higher and closer values to reality *LOD* values [30]. Since the mass spectrum of a given compound/peak can be compared with spectra of private or commercial spectrum libraries, the occurrence of a match at a given retention time ensures reliability of compound detection and identification. Additionally, considering the GC-MS method used, the *LOD* value will be related with the *LOQ* value as, if a given compound identification is reliable, its quantification from the chromatogram is possible. Hence, the statistical calculation of the *LOD* value has no practical application, and the experimental approach was performed for determining the *LOQ* value, which is of greater use. Several recommendations can be found for determining the *LOQ* value but considering the focus of the method. A conservative approach was chosen for its establishment by following the recommendations of Kruve and collaborators [25]. Therefore, the minimum amount of analyte detected and quantified was taken into account for determining *LOQ*, with the obtained values presented in Table 1. As demonstrated, *LOQ* values ranged from 2 µg/L to 7 µg/L for most compounds, with the exception of 3-methyl-1-pentanol (14.2 µg/L) for which the minimum tested concentration was higher, and ethyl lactate (107.4 µg/L) due to the lower sensitivity obtained. The obtained values are, in their majority, about 2 to 10 folds lower when compared with the values reported by Ferreira and collaborators [13], which worked with similar compounds and concentrations. As a cross validation for acceptance of this value, the measurements are within the 20% of the relative standard deviation (*RSD*), as stipulated by Brettel and Lester [30].

### 2.3. Precision

As reported by Kruve and collaborators [26], precision can be quantified as the relative standard deviation/coefficient of variation of replicate analysis. In this work, we evaluated two types of precision, the repeatability (a single operator in the same run conditions), and the intermediate precision (different operators, different run conditions but the same laboratory). The *RSD* values were calculated for the two scenarios which were presented in Table 2. For evaluating the intermediate precision, analyses using the proposed LLME method were performed by two operators with one experienced in its execution and one with reduced experience in the laboratory and with the method. Five replicates were measured by each operator using an independent equipment and apparatus, where the GC-MS is the only equipment in common for the analysis of extracts.

As visible in Table 2, *RSD* observed for evaluation of repeatability was considerably low, ranging from 3.3% to 9.0%. When analysing the *RSD* values obtained for intermediate precision (involving two different operators), a higher variation can be observed ranging from about 6.0% to 19.7%. This higher dispersion of the measurements can be justified by the differences in the experience of the operators, where deviations in the addition of an internal standard or differences in the interpretation and integration of chromatograms can lead to a higher dispersion of results. Establishment of critical *RSD* values for a method to be precise depends strongly on the application intended. Several limits have been proposed, which are the most common considered *RSD* < 15% of the nominal value [26,31] or as high as 20% for environmental or food samples [32]. As seen in the results, values of *RSD* regarding repeatability were all below the minimum level accepted. In addition, despite the higher *RSD* values obtained for intermediate precision, the majority of compounds were still below the acceptable limit of 15% with the exception of 3-methyl-1-pentanol, ethyl lactate, *E*-3-hexen-1-ol, and *Z*-3-hexen-1-ol, which still fall below the limit of 20% proposed by Huber [32]. Thus, the method is considered precise and can be performed with satisfactory outputs.

### 2.4. Accuracy

Accuracy was determined by the addition of a known amount of the analytes in the study to a real sample (spiking) and quantification of the analytes in the spiked sample. For this purpose, a commercial beer was analyzed using the proposed LLME method both in its original state and after spiking, as recommended by the guidelines for method validation [26]. To better assess accuracy, the theoretical expected concentration (*C*_expect_ = *C*_beer_ + *C*_spik_) was compared with the concentration measured using the LLME method (*C*_determin_). According to multiple *t*-tests for comparisons (*p* ≤ 0.05), no differences were found between the expected and the measured concentrations. As a more appropriate measure of accuracy, the deviation of the measured concentrations regarding the expected values was calculated, and expressed as a relative error (*RE*) [31]. In agreement with the results reported by González and collaborators [31], this value cannot exceed 15% for the method to be accurate (except for determinations at the *LOQ* where 20% is accepted). As shown in Table 2, *RE* values were within the 15% limit established for the studied compounds. Thus, the method is considered to be accurate when complying with another key requirement for validity.

### 2.5. Robustness

As stated, robustness can be defined as the ability of the method to endure slight variations and maintain its result [25]. To assess the robustness of the method, two criteria were evaluated including variation of contact time and the matrix effect, which were identified as the main variables affecting the LLME method. The effect of an increased stirring and extraction time was tested to evaluate the possible occurrence of differences in compound extraction. Again, as performed for accuracy, possible differences in compound recovery and quantification were statistically determined by the *t*-test, comparing the measured concentration with the expected concentration of the compounds, and evaluating the recovery of target analytes by taking into account the known dilution and concentration of the solution of standards.

Regarding the increase of stirring time, no statistically significant differences were observed for the measurements performed with 30 min of stirring (*p* > 0.05). Extraction of the compounds using 15 min stipulated in the method is shown to be sufficient for the total recuperation of analytes, which is maintained independently of the longer contact time. For a better assessment of robustness, recovery was calculated for each compound in accordance with Kruve and collaborators [26], with the values presented in Table 2. Similarly, when observed for accuracy, critical recovery values can be established for the acceptance of recovery in determining robustness. In this sense, values of recovery between 70% and 110% for measurements ranging from 10 µg/L to 100 µg/L, or 80% to 110% for measurements above 100 µg/L, are considered acceptable [33]. Therefore, the recovery values obtained with increased stirring time were within the acceptable range.

Regarding the matrix effect, the main focus was to evaluate if the recovery and quantification of the analyses would be affected by other components in the mixture. For control purposes, two synthetic matrixes were tested including a solution mimicking wine composition and another mimicking vinegar composition. For the majority of compounds, recovery values were also within the acceptable ranges previously referred, with the exception of those obtained for 3-methyl-1-pentanol and ethyl lactate in the synthetic wine matrix. The higher recovery observed for these compounds can be caused by a higher efficiency and selectivity in their extraction and, therefore, an accuracy test or validation in wine would be advised for the specific quantification of these compounds. Nevertheless, only two compounds in one matrix showed recovery values outside the acceptable range. The remaining compounds were properly quantified in the synthetic wine as well as all compounds in the synthetic vinegar matrix. Considering the overall results obtained under multiple conditions, global robustness of the method can be considered satisfactory.

## 3. Materials and Methods

### 3.1. LLME-GC/MS Method

#### 3.1.1. Liquid-Liquid Microextraction of Volatile Compounds

In a 10 mL culture tube (Pyrex, ref. 1636/26MP), 8 mL of sample, clarified by centrifugation if necessary, 2.46 µg of internal standard (4-nonanol, Merck ref. 818773), and a magnetic stir bar were added. Extraction was done by stirring samples with 400 µL of dichloromethane (Merck, ref. 106054), at room temperature for 15 min, using a magnetic stirrer. Tubes were placed vertically and agitation was regulated in order to maintain dispersion of solvent micro-drops without reaching the sample surface. After cooling at 0 °C for 10 min, the magnetic stir bar was removed and the organic phase was detached by centrifugation (5118 *g*, 5 min, 4 °C). Using a glass Pasteur pipette, the extract was recovered into a vial, dried with anhydrous sodium sulphate (Merck, ref. 1.06649), and transferred to a new vial for storage at –20 °C before analysis.

#### 3.1.2. Chromatographic Analysis

Gas chromatographic analysis of volatile compounds was performed using a GC-MS Varian Saturn 2000 (Varian, Walnut Creek, CA, USA) equipped with a 1079 injector, an ion-trap mass spectrometer, and a Sapiens-Wax MS capillary column (30 m × 0.15 mm, 0.15 µm film thickness, Teknokroma, Barcelona, Spain). The temperature of the injector and the MS transfer line were both set to 250 °C. The oven temperature was held at 60 °C, for 2 min, then programmed to rise from 60 °C to 234 °C, at 3 °C/min, and from 234 °C to 260 °C at 5 °C/min. Lastly, it was held for 10 min at 260 °C. The carrier gas was helium GHE4× (Praxair, Maia, Portugal), at a constant flow rate of 1.3 mL/min. A 1 µL injection was made in the split-less mode, for 30 s (split vent of 30 mL/min). The detector was set to an electronic impact mode (70 eV) with an acquisition range (*m*/*z*) from 35 to 300 at an acquisition rate of 610 ms.

#### 3.1.3. Identification of Volatile Compounds

Identification of volatile compounds was preformed using the software Star – Chromatography Workstation version 6.9.3 (Varian), by comparing mass spectra and retention indices with those of pure standard compounds.

### 3.2. Method Validation

#### 3.2.1. Base Standard Solution

To perform the method validation, a hydroalcoholic solution (7%, by volume; ethanol Fisher, 99.8%), using Milli-Q water, was initially prepared, which was the solvent used for compound dilution. First, a concentrated solution of the volatile compounds was prepared by adding each compound, by weighing using an analytical scale (Kern ABJ), to the hydroalcoholic solution, at a concentration of 1000× the highest concentration presented in Table 1. The base standard solution was then prepared by diluting the concentrated solution by a factor of 1000 with the hydroalcoholic solution to attain the highest concentrations specified in Table 1 (maximum value of the cited range). Compounds were chosen as being representative of the chemical groups with the higher impact in the volatile fraction and sensory properties of fermented beverages, such as wine, beer, and vinegar. These were purchased as pure standards with the purity and suppliers indicated in Table 1.

#### 3.2.2. Linearity

Calibration curves were constructed by using six points, corresponding to different concentrations obtained by the dilution of the base standard solution in the hydroalcoholic solution. Each solution was analyzed in triplicate by the proposed method. The average area ratios (*i.e.*, peak area of the compound x, $A_{x}$, to the peak area of the internal standard, $A_{\mathrm{IS}}$) were plotted against the concentration ratios (i.e., concentration of the compound x, $C_{x}$, to the concentration of the internal standard, $C_{\mathrm{IS}}$) to obtain the calibration curves in accordance with Equation (1).
(1)$$\frac{A_{x}}{A_{\mathrm{IS}}}=b\times \frac{C_{x}}{C_{\mathrm{IS}}}$$

From each curve, slope (*b*) and regression coefficient (*R*^2^) were calculated, and linearity was evaluated by a lack-of-fit *F*-test. Response factors ($R_{f}$) were also calculated for each compound as the inverse of the slope $\left(R_{f}=\frac{1}{b}\right)$. The limit of quantification (*LOQ*) was determined as the minimum concentration of the compound that could be trustily quantified.

#### 3.2.3. Precision

Two different measures of precision were evaluated for validation of the LLME method such as repeatability and intermediate precision. Repeatability was evaluated by the analysis of five replicate samples in the same conditions of the proposed method. As a measure of repeatability, the relative standard deviation (*RSD*) was calculated according to Equation (2).
(2)$$\frac{RSD}{\%}=\frac{s}{\bar{x}}\times 100$$
where s stands for standard deviation and $\;\bar{x}$ represents the average of the measured values. To evaluate intermediate precision, independent measurements of dissimilar samples were performed at different times by independent operators, where the *RSD* was also calculated as stated in Equation (2).

#### 3.2.4. Accuracy

In the absence of reference materials, accuracy was investigated by spiking and recovery. A commercial beer was used for analysis by the proposed method in its original state and after the addition (spiking) of a known mass of the analyte to the sample. The relative error (*RE*) of the determined concentration was calculated based on Equation (3), i.e., calculating the concentration of each compound in the spiked beer ($C_{\mathrm{determ}}$) against its expected concentration ($C_{\mathrm{expect}}$).
(3)$$\frac{RE}{\%}=\frac{C_{\mathrm{determ}}-C_{\mathrm{expect}}}{C_{\mathrm{expect}}}\times 100$$

#### 3.2.5. Robustness

Other parameters were studied to evaluate the susceptibility of the method to changes that might occur during routine analysis (use a different matrix or an extended extraction time). The matrix effect was evaluated using two different matrices mimicking a wine and a vinegar, respectively, by adding the same volatile compounds under evaluation at an intermediate concentration. Apart from volatile compounds, the synthetic wine comprises ethanol (12% by volume, Fisher, 99.8%), tartaric acid (5 g/L, Sigma, 99.5%), glycerol (7.5 g/L, Himedia, 99.5%), and malic acid (2 g/L, Acros Organics, 99%). Synthetic vinegar was prepared using 10 g/L of citric acid (Panreac, 99.5%) and 50 g/L of acetic acid (Sigma). Three replicates were carried out for each matrix. The effect of the change of the duration of the extraction time was also evaluated. Accordingly, three replicates of the extraction procedure were done by stirring the sample for 30 min instead of 15 min of the proposed method. Recovery (*Rec*) of target compounds, expressed as a percentage, was evaluated by calculating the measured concentration ($C_{\mathrm{measur}}$) vs. the expected concentration ($C_{\mathrm{expect}}$), as stated in Equation (4).
(4)$$\frac{Rec}{\%}=\frac{C_{\mathrm{measur}}}{C_{\mathrm{expect}}}\times 100$$

## 4. Conclusions

The LLME method presented in this work is a reliable alternative for the analysis of compounds participating in the volatile fraction of fermented beverages. The method is linear for the studied ranges and has good sensitivity, which varies depending on the chemical group of compounds. The method is precise and has shown good repeatability and intermediate precision. Variations were performed for the analytical matrix and for protocol execution. The LLME method was also demonstrated to be robust. Lastly, the method is accurate and adequate for application in real samples. Having complied with all the parameters needed, the LLME method presented in this work is, therefore, valid for application in the analysis of fermented beverages and, certainly, to distilled beverages/spirits, after a convenient dilution with water to reach an alcoholic strength, by volume, of about 15%.

## Figures and Tables

**Table 1 molecules-25-00621-t001:** Reference, purity (*P*), and concentration range (*C*) for each analyte, and Pearson correlation coefficient (*R*^2^), limit of quantification (*LOQ*), and response factor of the method ($R_{f}$), with respective confidence limits (*p* = 0.05), obtained from the calibration curves.

Compound	Reference	*P*/%	Range *C*/(µg/L)	*R* ^2^	*LOQ*/(µg/L)	$R_{f}$
4-methyl-2-pentanone	Fluka 02474	≥ 99.7	24.8 to 248	0.9991	6.9	1.32 ± 0.05
Ethyl butyrate	Aldrich E15701	99	5.76 to 576	0.9995	4.7	1.58 ± 0.04
Ethyl 2-methylbutyrate	Aldrich 306886	99	2.48 to 248	0.9997	1.8	0.87 ± 0.02
Ethyl 3-methylbutyrate	Aldrich 112283	98	3.12 to 312	0.9993	2.2	0.91 ± 0.03
3-methyl-1-butyl acetate	Aldrich 306967	≥ 99	21.32 to 2132	0.9990	3.9	2.00 ± 0.07
Ethyl hexanoate	Aldrich 148962	≥ 99	9.64 to 964	0.9978	2.2	1.32 ± 0.07
Hexyl acetate	Aldrich 108154	99	2.76 to 276	0.9983	2.9	1.57 ± 0.08
3-methyl-1-pentanol	Aldrich 111112	99	25.6 to 256	0.9968	14.2	4.63 ± 0.30
Ethyl lactate	Aldrich E34102	98	113.2 to 1132	0.9978	107.4	44.90 ± 2.45
1-hexanol	Fluka 73117	> 99.9	14.72 to 1472	0.9976	6.7	3.63 ± 0.20
*E*-3-hexen-1-ol	Aldrich 224715	97	6.32 to 632	0.9971	5.1	5.11 ± 0.32
*Z*-3-hexen-1-ol	Fluka 53056	≥ 98	7.20 to 720	0.9968	5.9	5.23 ± 0.34
Linalool	Aldrich L2602	97	4.76 to 476	0.9998	3.2	1.71 ± 0.03
Diethyl succinate	Aldrich 112402	99	6.12 to 612	0.9977	2.4	1.25 ± 0.07
α-terpineol	Merck 8.21078	≥ 98	2.60 to 260	0.9979	2.6	1.37 ± 0.07
Citronellol	Aldrich C83201	95	2.72 to 272	0.9999	2.2	1.43 ± 0.02
Nerol	Aldrich 268909	97	3.04 to 304	0.9988	3.1	1.83 ± 0.07
2-phenylethyl acetate	Fluka 46030	> 99	10.32 to 1032	0.9995	2.6	1.39 ± 0.03
Geraniol	Aldrich 163333	98	3.08 to 308	0.9994	2.4	1.26 ± 0.04
Guaiacol	Aldrich G10903	98	2.92 to 292	0.9984	5.1	2.65 ± 0.12
4-ethylphenol	Aldrich E44205	99	4.88 to 488	0.9983	4.2	2.03 ± 0.10

**Table 2 molecules-25-00621-t002:** Values obtained for evaluation of precision, measured as relative standard deviation (*RSD*), accuracy, expressed as relative error (*RE*), and robustness, quantified by compound recovery (*Rec*).

Compound	Repeatability	Intermediate Precision	Accuracy	Robustness		
	*RSD/%*	*RSD*/*%*	*RE*/%	*Rec*/% (*t* = 30 min)	*Rec*/% (Synthetic Wine)	*Rec*/% (Synthetic Vinegar)
4-methyl-2-pentanone	6.5	9.3	11.2	103.2	100.6	103.1
Ethyl butyrate	5.3	7.4	10.8	99.9	91.0	95.2
Ethyl 2-methylbutyrate	9.0	9.3	13.2	96.6	86.0	83.0
Ethyl 3-methylbutyrate	5.3	6.7	20.5	98.0	88.8	91.8
3-methyl-1-butyl acetate	4.5	5.7	1.6	104.7	93.9	96.2
Ethyl hexanoate	3.3	6.0	2.9	100.6	101.9	100.3
Hexyl acetate	3.8	11.2	15.5	98.0	92.3	97.8
3-methyl-1-pentanol	6.3	18.0	2.3	96.1	148.6	113.6
Ethyl lactate	8.4	18.9	2.8	88.5	161.6	106.7
1-hexanol	5.0	12.5	15.8	91.3	115.3	95.9
*E*-3-hexen-1-ol	7.0	18.0	14.2	86.0	116.3	96.2
*Z*-3-hexen-1-ol	6.8	19.7	15.9	85.8	116.4	90.5
Linalool	4.0	10.8	9.5	103.8	98.6	93.4
Diethyl succinate	3.3	10.3	13.3	113.0	117.7	113.4
α-terpineol	4.9	10.2	10.8	109.7	112.7	105.6
Citronellol	4.4	12.5	6.6	87.8	90.6	86.1
Nerol	5.9	13.8	13.7	108.6	100.0	93.2
2-phenylethyl acetate	3.4	7.9	0.7	108.6	109.6	109.9
Geraniol	2.6	9.6	16.7	107.2	97.3	93.9
Guaiacol	6.3	14.0	1.6	97.3	120.5	105.5
4-ethylphenol	4.5	9.9	5.0	72.9	112.5	97.1

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
