# Peer review of "Validation of a LLME/GC-MS Methodology for Quantification of Volatile Compounds in Fermented Beverages"

_molecules, 2020, doi:10.3390/molecules25030621_

Round 1

Reviewer 1 Report

The study by Coelho et al. presents the validation of a method for rapid analysis of a set of volatile compound in alcoholic berverages. The methodology is clear and well presented. The choices made are clearly argued and properly documented. I, therefore, recommend the work for publication.

Thouhg, I have minor comments to help the reader to understand the article.

l.22 - replace "LLME/GC-MS method is" > "LLME/GC-MS method may/must" : as no "real" sample is presented, there is no direct evidence that this method would work. The use of assertive sentence could lead the reader to the belief that it was tested for regular samples, which may have be done in your lab, but not presented (yet?) in the paper. A more speculative sentence is more appropriate.

l.86-89 "being recommended" is not correct here. The entire sentences are not clear. Many "being" are used. Please rephrase.

l.94 ... to the previously reported works for other...

At that point, and to clarify other points in the text, a graphe could be of use (regression, LOQ, Rt,...) presenting one or two characteristic compounds.

l.102 "the slope is presented in inverse, using the response factor Rf": that sentence does not make sense to me. I guess you mean that Rf is the inverse of the slope. Please rephrase.

l.120 Limits of detection (LOD) and quantification (LOQ) > define acronyms

l.125-129 : "Since the mass spectra... possible" > not clear, please rephrase.

l.128 "as, if" add a comma or the meaning would be different.

l.139 "RSD" define acronyms

Table 2 : be sure it appears on one page only.

l.167-169 : define "spiking" l.167 and not l.169

l.242 Could you explain more precisely the preparation of  base standard solution ? No information is given on the volatile compounds. Without giving the exact specification for all compounds, a few informations could be of use. Did you purchase them pure (what general purity) ? Do you produce many dilution to obtain the base satndard solution ? In which solvent ?  How ?

Author Response

Response to reviewers

In this document we present the comments made by the reviewers accompanied by a response (marked in blue font) as well as a transcript of the modified elements in the manuscript (marked in italic green font) where such is justifiable.

The study by Coelho et al. presents the validation of a method for rapid analysis of a set of volatile compound in alcoholic beverages. The methodology is clear and well presented. The choices made are clearly argued and properly documented. I, therefore, recommend the work for publication. Though, I have minor comments to help the reader to understand the article.

LINE 22 - replace "LLME/GC-MS method is" > "LLME/GC-MS method may/must" : as no "real" sample is presented, there is no direct evidence that this method would work. The use of assertive sentence could lead the reader to the belief that it was tested for regular samples, which may have be done in your lab, but not presented (yet?) in the paper. A more speculative sentence is more appropriate.

Modified as suggested.

lLINE 86-89 "being recommended" is not correct here. The entire sentences are not clear. Many "being" are used. Please rephrase.

Modified to address the reviewer suggestion.

“R2 value is a useful indicator of the regression quality but, according to Kruve and collaborators [26] and Araújo [28], it cannot be considered as a standalone measure to validate a method linearity, which must be further validated by a statistical lack-of-fit Ftest.”

LINE 94 ... to the previously reported works for other...

Modified accordingly.

At that point, and to clarify other points in the text, a graphe could be of use (regression, LOQ, Rt,...) presenting one or two characteristic compounds.

A graphic of a standard calibration curve was presented in the graphical abstract to provide such example.

LINE 102 "the slope is presented in inverse, using the response factor Rf": that sentence does not make sense to me. I guess you mean that Rf is the inverse of the slope. Please rephrase.

Rephrased as suggested.

LINE 120. Limits of detection (LOD) and quantification (LOQ) > define acronyms

Included as suggested.

LINE 125-129 : "Since the mass spectra... possible" > not clear, please rephrase.

The sentence was rephrased for better nderstanding.

"Since the mass spectrum of a given compound/peak can be compared with spectra of private or commercial spectrum libraries, the occurrence of a match at a given retention time ensures reliability of compound detection and identification."

LINE 128 "as, if" add a comma or the meaning would be different.

Modified accordingly.

LINE 139 "RSD" define acronyms.

Acronym defined.

Table 2 : be sure it appears on one page only.

We modified the supplied version of the manuscript to place Table 2 in on one page only.

This can be altered during editing and will be checked in the Proof version if the manuscript is accepted for publication.

LINE 167-169 : define "spiking" lLINE 167 and not LINE169

Modified accordingly.

LINE 242 Could you explain more precisely the preparation of base standard solution ? No information is given on the volatile compounds. Without giving the exact specification
for all compounds, a few informations could be of use. Did you purchase them pure (what general purity) ? Do you produce many dilution to obtain the base satndard solution ? In which solvent ? How ?

More details regarding preparation of the base standard solution were included in section 3.2.1, for a better comprehension of the methodology used.

“To perform the method validation, a hydroalcoholic solution (7%, by volume; ethanol Fisher, 99.8%), using Milli-Q water, was initially prepared, which was the solvent used for compound dilution. Firstly, a concentrated solution of the volatile compounds was prepared by adding each compound, by weighing using an analytical scale (Kern ABJ), to the hydroalcoholic solution, at a concentration 1000× the highest concentration presented in Table 1. The base standard solution was then prepared by diluting the concentrated solution by a factor of 1000 with the hydroalcoholic solution to attain the highest concentrations specified in Table 1 (maximum value of the cited range). Compounds were chosen as being representative of the chemical groups with the higher impact in the volatile fraction and sensory properties of fermented beverages, namely wine, beer and vinegar. These were purchased as pure standards, with purity and suppliers indicated in Table 1.”

Reviewer 2 Report

General: The manuscript is technically correct.  However, it packaged with notions of being ‘simple’, ‘fast’, ‘cheap’, and ‘less pollutant’, which are never substantiated with a reasonable metric of ‘simplicity’, a measure of time, cost per sample, ‘pollutant’ reduction.  Those metrics should be then presented in the context of current methods for the same class of compounds and matrixes. Also, the manuscript lacks a clear presentation of novelty and significance.  The LLME (ref. 27) was published in 2006.  Readers would appreciate knowing what the specific gap in knowledge that has not been addressed in the last 14 years is.

Title: The title can be misleading, specifically the “Simple and Quick” needs to be substantiated in comparison to other techniques.  Also, what might be ‘quick’ today will be very slow with future technologies.  I would encourage the Authors to reflect on this and provide additional rubric/analysis comparing ‘simplicity’ and ‘time’. A more adequate title might be a ‘method validation’. 

Abstract:

L14 – replace ‘a previous’ with ‘as a previous’.

L14-15 – the ‘cheap, fast and simple’ needs to be substantiated with numbers.  It is difficult to comprehend how a very sophisticated analytical platform (GC-MS) can be ‘cheap’ and an LLME be ‘simple’ and ‘fast’ without robotics and software, magnetic stir bars, Milli-Q water, -20 deg C storage. Some samples may require centrifugation. So it might be linguistics that is correct only for the specialized analytical laboratories with a sizeable budget to afford GC-MS with the robotic arm and trained staff. Many laboratories around the world cannot afford it.

L19-20 – please provide some numeric values (including statistics)

L23 – ‘less pollutant’ – please add some percentage of pollutant generation. 

Introduction:

It needs to be improved to specifically describe the problem/need and then very clearly state why the LLME can address the problem. It needs to provide a reader with a description of novelty and significance. The statement of objectives (L65-66) does not provide a reader with any specific information about the ‘validation’, reasons for choosing LLME. L67-69 – “This procedure….is scheduled to be applied in every laboratory equipped with a GC-MS, by any technician, …” – this is a gross overstatement. The word ‘is scheduled’ needs to be revised to, e.g., ‘could be used in laboratories…” L71 – replace ‘amount’ with ‘number’.

Results:

Table 1 – please provide a sentence or two describing the rationale for choosing these compounds.

Please add some comparisons of the LODs with other methods for the same classes of compounds in these matrixes.

Methods:

Please provide a figure schematic for LLME and some key sample preparation steps to help readers understand how ‘simple’ this method could be. 

Conclusions:

The statements of ‘fast, simple and cost-effective alternative’ do not have support in the presented data.  There are no comparisons of time, simplicity, and cost to established methods.

Author Response

Response to reviewers

In this document we present the comments made by the reviewers accompanied by a response (marked in blue font) as well as a transcript of the modified elements in the manuscript (marked in italic green font) where such is justifiable.

General: The manuscript is technically correct. However, it packaged with notions of being ‘simple’, ‘fast’, ‘cheap’, and ‘less pollutant’, which are never substantiated with a reasonable metric of ‘simplicity’, a measure of time, cost per sample, ‘pollutant’ reduction. Those metrics should be then presented in the context of current methods for the same class of compounds and matrixes. Also, the manuscript lacks a clear presentation of novelty and significance.
The LLME (ref. 27) was published in 2006. Readers would appreciate knowing what the specific gap in knowledge that has not been addressed in the last 14 years is.

The LLME methodology published in 2006 was used for analysis of only three compounds, all C6-alcohols, namely 1-hexanol, E-3-hexenol and Z-3-hexenol. Also, these compounds were only analyzed in wine. As the method provided satisfactory results, the presented work performed its validation for the analysis of a broader range of compounds, beyond the ones initially tested. To make this clearer to the readership, introduction was modified to better highlight the incremental value and novelty of the presented work.

“This work aims at the validation of a liquid-liquid microextraction method (LLME) first published by Oliveira and collaborators [27], which only reported its use for the analysis of three C6-alcohols (1-hexanol, E-3-hexenol and Z-3-hexenol), exclusively in wine. As the method provided satisfactory performance and results, its feasibility for the analysis of a broader range of compounds and a wider variety of matrices remained to be validated.”

Title: The title can be misleading, specifically the “Simple and Quick” needs to be
substantiated in comparison to other techniques. Also, what might be ‘quick’ today will be very slow with future technologies. I would encourage the Authors to reflect on this and provide additional rubric/analysis comparing ‘simplicity’ and ‘time’. A more adequate title might be a ‘method validation’.

We acknowledge that the consideration of “Simple and Quick” is presented in a subjective manner and that it can be relative in terms of the available technology. In order to avoid any confusion, we removed these claims from the manuscript as they could be in some manner misleading. Therefore, the title was modified to address these concerns.

Validation of a LLME/GC-MS Methodology for Quantification of Volatile Compounds in Fermented Beverages.

Abstract:
L14 – replace ‘a previous’ with ‘as a previous’.

Modified accordingly.

L14-15 – the ‘cheap, fast and simple’ needs to be substantiated with numbers. It is difficult to comprehend how a very sophisticated analytical platform (GC-MS) can be ‘cheap’ and an LLME be ‘simple’ and ‘fast’ without robotics and software, magnetic stir
bars, Milli-Q water, -20 deg C storage. Some samples may require centrifugation. So it
might be linguistics that is correct only for the specialized analytical laboratories with a
sizeable budget to afford GC-MS with the robotic arm and trained staff. Many
laboratories around the world cannot afford it.

Again, we acknowledge that the concepts are subjective and strongly dependent on the reality of each laboratory and the available technology. Thus, we modified the manuscript to eliminate such subjective statements.

L19-20 – please provide some numeric values (including statistics)
L23 – ‘less pollutant’ – please add some percentage of pollutant generation.

As “less pollutant” was a dubious concept, it was removed from the manuscript.

Introduction:
It needs to be improved to specifically describe the problem/need and then very clearly state why the LLME can address the problem. It needs to provide a reader with a description of novelty and significance. The statement of objectives (L65-66) does not provide a reader with any specific information about the ‘validation’, reasons for choosing LLME.

These concerns were addressed in the previous response and the statement of objectives was modified to better demonstrate the motivation behind this work taking into consideration the state of the art.

“This work aims at the validation of a liquid-liquid microextraction method (LLME) first published by Oliveira and collaborators [27], which only reported its use for the analysis of three C6-alcohols (1-hexanol, E-3-hexenol and Z-3-hexenol), exclusively in wine. As
the method provided satisfactory performance and results, its feasibility for the analysis of a broader range of compounds and a wider variety of matrices remained to be validated.”

L67-69 – “This procedure….is scheduled to be applied in every laboratory equipped with a GC-MS, by any technician, …” – this is a gross overstatement. The word ‘is scheduled’ needs to be revised to, e.g., ‘could be used in laboratories…”

Sentence was modified to address the reviewer's suggestion.

“This procedure can be applied in any laboratory equipped with a GC-MS, by any
technician, using only ordinary glassware and low amounts of sample and solvents.”

L71 – replace ‘amount’ with ‘number’.

Modified accordingly.

Results:
Table 1 – please provide a sentence or two describing the rationale for choosing these compounds.

A sentence justifying the rationale is present in the results section (L79-L82)

“…solution of pure standards. Compounds selected for calibration of the method were chosen on the basis of their contribution for the volatile and aromatic fraction of fermented products, being considered to be representative of the analytes generally found in beer, wine, spirits and vinegar.”

Please add some comparisons of the LODs with other methods for the same classes of compounds in these matrixes.

A comment comparing LODs with other methods was included as suggested.

“The obtained values are, in their majority, about 2 to 10 folds lower when compared with the values reported by Ferreira and collaborators [13], which worked with similar compounds and concentrations.”

Methods:
Please provide a figure schematic for LLME and some key sample preparation steps to help readers understand how ‘simple’ this method could be.

We didn’t include a figure schematic of the LLME protocol because it is described in section 3.1.1 with extensive detail. Also, a figure schematic depicting the microextraction phase, which is the key stage of the method, was already constructed and supplied in the graphical abstract. To our consideration, a figure schematic depicting the remaining
protocol, which only includes basic laboratory manipulation steps, wouldn’t provide any additional detail.

Conclusions:
The statements of ‘fast, simple and cost-effective alternative’ do not have support in the presented data. There are no comparisons of time, simplicity, and cost to established methods.

As referred in previous responses, such subjective statements were removed from the manuscript.

Reviewer 3 Report

The paper reports the validation of an existing analytical method developed and published in 2006 by Oliveira et al. Methodology is well described and confirms the method as robust, reproducible and precise.

However, in my opinion, this doesn’t justify a new publication because:

Method is the same as reported in the original paper (e.g. paragraph 3.1.1. is practically identical) Instrumentation adopted is the same Only a reference solution in alcohol and water was considered, while to confirm the validity of the method it is necessary, in my opinion, to consider also real world samples.

Due to the previous considerations, it is not clear how this paper improve the methodologies to be adopted for analysis of volatile compounds in alcoholic beverages

Author Response

In this document we present the comments made by the reviewers accompanied by a response (marked in blue font) as well as a transcript of the modified elements in the manuscript (marked in italic green font) where such is justifiable.

The paper reports the validation of an existing analytical method developed and published in 2006 by Oliveira et al. The methodology is well described and confirms the method as robust, reproducible and precise.

However, in my opinion, this doesn’t justify a new publication because:

Method is the same as reported in the original paper (e.g. paragraph 3.1.1. is practically identical). Instrumentation adopted is the same. Due to the previous considerations, it is not clear how this paper improve the methodologies to be adopted for analysis of volatile compounds in alcoholic beverages.

The method reported in the original paper was only used for the analysis of three C6-alcohols, exclusively in wine matrices. The feasibility of the LLME method for the analysis of other groups of compounds and applicability to other fermented matrices remained to be validated, which was the aim of the presented work. We acknowledge that this wasn’t clear in the introduction and modified it accordingly to better reflect the incremental value and novelty of this work.

“This work aims at the validation of a liquid-liquid microextraction method (LLME) first published by Oliveira and collaborators [27], which only reported its use for the analysis of three C6-alcohols (1-hexanol, E-3-hexenol and Z-3-hexenol), exclusively in wine. As the method provided satisfactory performance and results, its feasibility for the analysis of a broader range of compounds and a wider variety of matrices remained to be validated.”

Only a reference solution in alcohol and water was considered, while to confirm the validity of the method it is necessary, in my opinion, to consider also real world samples.

Real world samples were also considered in the presented work, as accuracy determinations were performed using real beer spiked with the studied compounds.

Round 2

Reviewer 2 Report

The authors addressed my comments in R1. Additional editing

Reviewer 3 Report

The authors have modified the introduction explaining the meaning of the work, in particular evidencing the improvement versus the previous papers published. In my opinion the paper can be now accepted for publication.